# Enhanced Sensitivity of Electrochemical Sensors for Ammonia-Nitrogen via In-Situ Synthesis PtNi Nanoleaves on Carbon Cloth

**DOI:** 10.3390/s24020387

**Published:** 2024-01-09

**Authors:** Guanda Wang, Guoxing Ma, Jie Gao, Dong He, Chun Zhao, Hui Suo

**Affiliations:** State Key Laboratory of Integrated Optoelectronics, College of Electronic Science and Engineering, Jilin University, Changchun 130012, China; gdwang19@mails.jlu.edu.cn (G.W.); mgx4101616@163.com (G.M.); gao1229jie@163.com (J.G.); hedong@jlu.edu.cn (D.H.); zchun@jlu.edu.cn (C.Z.)

**Keywords:** electrochemical sensor, PtNi alloy nanoleaves, ammonia-nitrogen detection, enhanced sensitivity

## Abstract

Pt-based electrochemical ammonia-nitrogen sensors played a significance role in real-time monitoring the ammonia-nitrogen concentration. The alloying of Pt and transition metals was one of the effective ways to increase the detectability of the sensitive electrode. In this paper, a self-supported electrochemical electrode for the detection of ammonia nitrogen was obtained by the electrodeposition of PtNi alloy nanoleaves on a carbon cloth (PtNi-CC). Experimental results showed that the PtNi-CC electrode exhibited enhanced detection performance with a wide linear range from 0.5 to 500 µM, high sensitivity (7.83 µA µM^−1^ cm^−2^ from 0.5 to 150 μM and 0.945 µA µM^−1^ cm^−2^ from 150 to 500 μM) and lower detection limit (24 nM). The synergistic effect between Pt and Ni and the smaller lattice spacing of the PtNi alloy were the main reasons for the excellent performance of the electrode. This work showed the great potential of Pt-based alloy electrodes for the detection of ammonia-nitrogen.

## 1. Introduction

Ammonia nitrogen refers to nitrogen in water in the form of free ammonia (NH_3, aq_) and ammonium ions (NH_4_^+^) [1]. Ammonia nitrogen was one of the important water quality indexes in water, which is usually used to evaluate the degree of decomposition of organic matter in water and the nitrogen load in water. Excessive ammonia nitrogen concentration will lead to the eutrophication of water, cause algae growth, and destroy the self-purification ability and ecological balance of water [2]. When ammonia nitrogen enters the human water supply, it reacts with chlorine-containing disinfectants to produce cancer-causing nitrosamines. For aquatic crustaceans, ammonia nitrogen is also a deadly neurotoxin, and its concentration can cause huge economic losses in the aquaculture industry [3]. Therefore, it is necessary to establish an ammonia-nitrogen detection system to monitor the concentration of ammonia in water in real time [4].

At present, the traditional analysis methods of ammonia nitrogen include Nessler’s reagent method, the distillation titration method and spectrophotometry [5,6,7,8]. Although the above methods had excellent selectivity and sensitivity, they also had disadvantages such as cumbersome operation and long time, so that they are not suitable for the rapid and real-time monitoring of ammonia nitrogen [9]. Therefore, electrochemical methods with simple operation, fast detection and high sensitivity have gradually attracted the attention of researchers [10]. At present, researchers are constantly working to improve the sensitivity, selectivity and stability of electrochemical detection methods for ammonia nitrogen in order to better meet the needs of practical applications [11]. As the core detection element, the electrochemical ammonia-nitrogen sensitive electrode has become the key research object [12].

Ammonia-sensitive electrode materials mainly include precious metal nanomaterials, transition metals and their oxides and conductive polymers. Researchers have conducted many studies on these sensitive materials. Novell-Leruth et al. calculated the adsorption characteristics of Pt(100) and Pt(111) crystal faces for ammonia by using DFT theory [13]. Endo et al. prepared Pt-Ir and Pt-Cu binary alloys as electrocatalysts for ammonia oxidation [14]. They proved that a synergistic interaction between Pt and Ir can enhance the ammonia oxidation activity. Yang et al. implemented 3D Rosett-like Cu nanoparticles with a sensitive detection of ammonia nitrogen [15]. Jiang et al. studied the effect of Zn modification on the catalytic activity of PtIr and Pt for ammonia electrooxidation [16]. In many of the studied species, noble metal nanomaterials, especially platinum, show excellent sensitivity to ammonia nitrogen. Although transition metals can also realize better ammonia-sensitive properties, they were often used as modified materials and combined with precious metal materials to obtain better performance-sensitive electrode materials. However, Pt-based sensors must face two problems in the process of practical application. On the one hand, the price of Pt is expensive, which will lead to higher sensor costs; on the other hand, in the process of detecting ammonia, the intermediate products can easily to poison the Pt, which affects the service life of the sensor [17]. Therefore, reducing the content of Pt in the sensor and improving its sensitivity to ammonia nitrogen are the key to further developing electrochemical ammonia-nitrogen sensors.

At present, transition metal alloying is an effective way to solve these problems, and this method has shown excellent results in different kinds of fields, like hydrogen evolution reaction (HER), oxygen reduction reaction (ORR), and ammonia oxidation reaction (AOR) [18,19,20]. Zhang et al. constructed the surface-structure tailoring of dendritic PtCo nanowires for efficient oxygen reduction reaction. Tran et al. graphene-coated PtNi nanosponges to enhance the oxygen reduction performance. Some studies have shown that changing the Pt-Pt bond spacing by alloying can affect the electronic structure of Pt and improve the electrocatalytic oxidation ability of Pt-based nanomaterials to ammonia. In addition, the Density Functional Theory (DFT) calculation shows that the transition metal has weak adsorption ability to the intermediate products of ammonia oxidation [21,22], so Pt-M (M can be other transition metals, like iron, nickel, cobalt, copper, etc.) nanomaterials have better toxicity resistance [19,23,24,25]. Among the transition metal elements, the Ni element has a small atomic radius and a good ability to catalyze ammonia, and it has a certain toxicity resistance to ammonia [26], so it is one of the more ideal elements for alloying with Pt.

In most sensitive electrodes, the sensitive material is fixed on the collector liquid by an adhesive, which will inevitably lead to a reduction in the activity of the sensitive material [27]. In contrast, sensitive electrodes with self-supporting structures have an advantage in retaining material activity. Therefore, the design for sensors with self-supporting structures could improve the specific surface area, which can supply more active sites, enhancing the sensitivity to the detected object. Carbon cloth (CC) is one of the commonly used collector electrodes because of the excellent corrosion resistance to acid and alkali, as well as its outstanding electrical conductivity [15].

In this paper, we use a one-step electrodeposition in situ growth strategy to synthesize platinum nickel alloy nanomaterials (PtNi) on the carbon cloth collector electrode and prepare electrochemical ammonia-nitrogen-sensitive electrodes with a self-supporting electrode structure. Due to the successful alloying of nickel atoms with platinum, the electrode exhibits an enhanced electrochemical detection capability of ammonia nitrogen with lower cost and higher stability. The self-supporting electrode structure also avoids an unnecessary loss of active sites caused by binders. The electrochemical sensitivity of the electrode to ammonia was evaluated. The excellent performance of the PtNi alloy nanomaterial electrode for ammonia-nitrogen detection indicates that the development of a modified electrode based on PtNi alloy is an effective strategy to enhance the detection of ammonia nitrogen.

## 2. Materials and Methods

### 2.1. Reagents and Chemicals

Ultrapure water (18.2 MΩ·cm), chloroplatinic acid hexahydrate (H_2_PtCl_6_·6H_2_O, AR), ammonium chloride (NH_4_Cl, AR), ammonium fluoride (NH_4_F, AR), potassium hydroxide (KOH, AR), hydrochloric acid (HCl, AR), sodium chloride (NaCl, AR), potassium carbonate (K_2_CO_3_, AR), sodium bicarbonate (NaHCO_3_, AR), and nickel chloride (NiCl_2_·6H_2_O, AR) were used as received. All reagents are from Beijing Shiji Company (Beijing, China).

### 2.2. Apparatus and Equipment

Field-emission scanning electron microscope (SEM) was used to investigate the morphologies of samples (JEOL-JEM-6700F, Tokyo, Japan). Transmission Electron Microscopy (TEM, JEOL JSM-7500 F) with energy-dispersive X-ray spectroscopy (EDS, JEOL Japan) was used to confirm the microstructure of samples. The structures of samples were characterized by X-ray diffraction (XRD, Shimadzhu-6000, Cu Kα radiation, λ = 1.54056 Å) and X-ray photoelectron spectroscopy (XPS, ESCALAB-250). The synthesis of samples and electrochemical experiments was performed on the electrochemical workstation (CHI 760D, Chenhua, Shanghai, China) with the typical three-electrode system. A Pt sheet (1.5 cm × 1.5 cm) and Hg/HgO electrode act as the counter-electrode and reference electrode, respectively. The cyclic voltammetry measured (CV) on PtNi-CC was from −0.8 to 0.2 V at scanning rate of 50 mV s^−1^. The differential pulse voltammetry (DPV) measured on PtNi-CC was from −0.55 to −0.15 V at a pulse amplitude of 50 mV, pulse width of 0.2 s and pulse period of 0.5 s. The electrolyte used for the electrochemical test was 1 M KOH solution.

### 2.3. Synthesis of PtNi-CC Electrode

Firstly, the carbon cloth was cut into rectangles of 1 cm × 1 cm. Then, the CC electrodes were washed by toluene, acetone, ethanol and hydrochloric acid for several minutes; then, they were rinsed by deionized water and dried at 60 °C for 2 h. 

A three-electrode system was used. A carbon cloth, platinum sheet (1.5 cm × 1.5 cm) and mercury oxide reference electrode (Hg/HgO) were used as the working electrode, counting electrode and reference electrode, respectively. In the solution containing NiCl_2_·6H_2_O (0.6 mM) and H_2_PtCl_6_·6H_2_O (2.4 mM), 25 cycles of Pt nanoparticles were electrodeposited in the range of −0.8 to 0.6 V (VS Hg/HgO) at a scanning rate of 0.05 V s^−1^ by the cyclic voltammetry (CV) method. The final product was washed with deionized water for several times and was named Pt4Ni1-CC. As control experiments, the samples synthesized under the different Pt and Ni ratios in the precursor electrolyte with 3:0, 4:1, 3:2, 2:3, 1:4 and 0:3 were donated as Pt-CC, Pt4Ni1-CC, Pt3Ni2-CC, Pt2Ni3-CC, Pt1Ni4-CC, and Ni-CC, respectively.

## 3. Results and Discussion

### 3.1. Material Characterization

Figure 1 shows the SEM images of sensitive electrodes loaded with different nanomaterials. Under the condition of high deposition overpotential, the Pt ion was reduced, and the Pt nucleus grew in the direction of the two-dimensional structure and formed a leaf shape. As can be seen from the image in Figure 1a_1_, the Pt nanoleaves were stacked in a disordered manner on the surface of the CC. When Ni ions were added to the electrodeposition precursor, the length of the Pt nanosheets began to decrease (Figure 1b_1_). At this time, the image at low magnification (Figure 1b_2_) shows that the leaf-like nanosheets were combined into distinct flower clusters of nanomaterials. With the further increase in Ni atoms, the deposited nanoleaves gradually resembled nanoblocks or even irregular nanospheres (Figure 1c_1_–e_1_). At low magnification, the nanomaterials of the flower clusters gradually became smaller and eventually transformed into tiny particles (Figure 1b_2_–e_2_), which also corresponded to the morphology of the high magnification images. As for the Ni-CC electrode, the nickel nanoparticles appeared in the form of an irregular blocky structure and were distributed unevenly on the surface of the CC electrodes. We suspect that the growth of Pt nanocrystals was affected by Ni atoms during electrodeposition.

The microstructure of the Pt4Ni1-CC electrode was analyzed by the high-magnification TEM and the high-resolution transmission electron microscope (HR-TEM). From the images, it can be seen the lattice fringes of PtNi nanoleaves were clear and consistent in direction. In Figure 2a, the calculated crystal face with a lattice spacing of 0.220 nm corresponds to the (111) crystal face of the PtNi alloy [28]. To further determine the coexistence of Pt atoms and Ni atoms, the element mapping of PtNi alloys was analyzed by high-angle toroidal dark-field scanning TEM energy dispersive X-ray spectroscopy (HAADF-STEM-EDS). As shown in Figure 2b–e, PtNi nanomaterials were composed of Pt and Ni elements. And Pt and Ni atoms were evenly distributed in the sample and formed the uniform alloy structure [26,29]. The presence of Pt and Ni elements was quantitatively determined by EDS data. The Pt to Ni atomic ratio of Pt4Ni1-CC sample is 68.11:31.89.

To confirm the crystal structure of the Pt and PtNi alloy, Pt-CC, Pt4Ni1-CC, Pt3Ni2-CC, Pt2Ni3-CC and Pt1Ni4-CC were characterized by XRD. As shown in Figure 3a, there were five characteristic peaks at 39.97°, 46.25°, 67.48°, 81.48° and 85.81° in the XRD pattern of Pt-CC. These five peaks corresponded to the Pt (111), (200), (220), (311) and (222) crystal planes of fcc structure type (Pt-PDF#04–0802). It was indicated that the Pt nanoleaves were electrodeposited on the CC electrode. As the content of Ni atoms increases, the 2θ angle of the Pt (111) peak was observed to shift in a larger direction (Figure 3b, indicating that the lattice of PtNi alloy shrank slightly compared to pure Pt. In addition, as shown in Figure 3a, there were no diffraction peaks of Ni or nickel oxides that appeared on the spectrum of each sample.

Since the atomic radius of Ni was smaller than that of Pt, the lattice of Pt shrank when Pt atoms were alloyed with Ni, so the XRD peaks moved to a larger angle direction. Meanwhile, the peaks’ intensity of different PtNi alloy electrodes also decreased in the XRD patterns when the content of nickel atoms increased. The results showed that the crystallinity of platinum decreased. This view can be verified with the morphological changes of Pt and PtNi nanoleaves in Figure 1a_1_–e_1_ [30,31].

Until now, it had been generally accepted that modifications of platinum’s electronic structure played a crucial part in enhancing activity. For further determining the electronic structure of Pt, XPS was used to analyze Pt-CC and Pt4Ni1-CC. The full scan spectrum is shown in Figure 3c, which contains all the element peaks including Pt, C, O, and Ni. Figure 3d shows the Pt 4f binding energy region of two catalysts (Pt and PtNi). As shown in Figure 3d, the Pt 4f spectra were deconvolved into two peaks corresponding to ${4f}_{7/2}$ and ${4f}_{5/2\;}$. According to the literature, energy separation was 3.3 eV [28]. The Pt ${4f}_{7/2}$ and ${4f}_{5/2\;}$ for PtNi alloy nanoleaves were shifted at 0.5 eV compared with the peak of Pt. The shift of Pt 4f (0.5 eV) was caused by the transfer of the electrons from the low electronegative Ni to Pt, which indicated that the PtNi alloy was successfully formed [32].

Table 1 lists the banding energy values of Pt-CC and Pt4Ni1-CC. The two characteristic peaks of Pt from the Pt4Ni1-CC electrode were at 71.70 eV and 75.00 eV, respectively, which belonged to Pt(0). And the two characteristic peaks corresponding to Pt(II) were located at 72.81 eV and 76.11 eV, respectively. 

Clearly, the content of Pt(0) from the PtNi alloy nanoleaves was lifted obviously by comparison with the relative intensities of Pt(0) between Pt and PtNi in Table 1 (from 45.67% to 62.50%). It indicated the addition of Ni can inhibit the forming of platinum oxide. Because Pt(0) provided more active sites during the process of ammonia oxidation, Pt4Ni1-CC can exhibit enhanced detectability. It also explains that the binding energy of Pt(0) 4f from PtNi alloy nanoparticles (71.70 eV) increased by 0.5 eV compared with Pt(0) 4f of pure Pt nanoparticles (71.20 eV). This is like that of the alloyed Pt-based materials reported in the literature [19,28].

### 3.2. Study on Electrochemical Sensitivity Characteristics

The sensitivity of different kinds of electrodes to ammonia are analyzed in Figure 4a–e. It shows that the CV curves of different kinds of electrodes included Pt foil, Pt-CC, Pt4Ni1-CC, Ni-CC and pure CC in the electrolytes with and without 5 mM NH_4_Cl solution. Obviously, the CC electrode has almost no electrochemical response to ammonia. And the performance of the Ni-CC electrode to ammonia is also much smaller than that of other platinum-based electrodes. The specific response current value can be obtained from Figure 4f.

As shown in Figure 4a–e, except for the CC electrode, all the other electrodes had similar peaks in the CV cures. The difference was that the two pairs of distinct peaks between −0.8 and −0.55 V were due to the process of hydrogen adsorption/desorption [33]. And the peaks between −0.25 and −0.4 V in the negative scanning curves were the adsorption of OH^−^ [30]. It can be seen from Figure 4b that the CV curve of Pt4Ni1-CC showed stronger peaks than the Pt-CC electrode and commercial Pt foil electrode, which indicated that more active sites appeared on the Pt4Ni1-CC electrode surface.

As shown in Figure 4a–d, the curves of all electrodes (Pt foil, Pt-CC, Pt4Ni1-CC and Ni-CC) showed oxidation peaks at around −0.28 V, which were the peaks of ammonia electrocatalytic oxidation. It indicated that these electrodes possessed electrocatalytic activity to ammonia, and the Pt4Ni1-CC electrode had performed the best, as shown in Figure 4f. Based on previous reports of Pt-based electrocatalysts for ammonia oxidation, ammonia is catalyzed to molecular nitrogen (N_2_) [33]. The mechanism of ammonia electrooxidation had been sufficiently reviewed in past studies, and a widely accepted theory was proposed by Gerischer and Mauerer [34]. From Equations (1) and (2), the Pt-based electrocatalyst dehydrogenated the adsorbed NH_3_ (NH_3,aq_) continuously, generated various active intermediates and finally generated N_2_ [34].

The overall anode reaction is shown below:NH_3,aq_ → NH_3,ads_(1)
2NH_3(aq)_ → N_2,g_ + 6H^+^ + 6 e^−^(2)

According to the past studies, the decrease in crystal surface spacing after Pt and transition metal alloying was the reason for the enhancement to ammonia electrocatalytic oxidation ability. Therefore, the sensitivity of the Pt4Ni1-CC electrode to ammonia was higher than that of the Pt-CC electrode obtained by the same preparation method. In addition, the synergistic effect between Pt and Ni atoms may be another significant factor in improving the electrochemical responding of the alloyed electrodes to ammonia. According to the above XRD patterns, the crystal plane spacing of Pt decreased after the formation of the PtNi alloy, which was conducive to improving the electrochemical activity of ammonia in the electrooxidation process and improving the response current.

The effect of different element ratios on the PtNi-CC electrode was evaluated. Pt-CC, Pt4Ni1-CC, Pt3Ni2-CC, Pt2Ni3-CC, and Pt1Ni4-CC electrodes were tested in the electrolyte with 5 mM ammonia (Figure 5a,b). As can be seen from Figure 5b, the responding peak current value of the Pt4Ni1-CC electrode was significantly higher than those of the others. The peak current density of the Pt4Ni1-CC electrode was 3.795 mA cm^−2^, which was higher than those of Pt-CC (2.805 mA cm^−2^), Pt3Ni2-CC (3.430 mA cm^−2^), Pt2Ni3-CC (2.006 mA cm^−2^) and Pt1Ni4-CC (1.070 mA cm^−2^). This showed that the sensitivity of Pt-based electrodes for ammonia can be improved by adding a certain proportion of Ni atoms, but this ability gradually decreased with the further increase in the proportion of Ni atoms. 

It is very important to study the electron transport kinetics and interfacial electrochemical properties of the modified electrode. As described in Figure 5c, all the electrodes revealed reversible redox peaks (I_PtNi-CC_ > I_Pt-CC_ > I_Ni-CC_ > I_CC_). The peak current of PtNi-CC was significantly higher than that of other electrodes. This showed that the electrode obtained a larger active specific surface area and higher conductivity by electrodepositing PtNi nanoleaves on the bare CC surface.

Figure 5d depicted the Nyquist plots of PtNi-CC, Pt-CC, PtNi-CC, PtNi-CC and bare Pt foil electrodes, and the illustration was the partial enlarged drawing. The Nyquist plots consisted of a semicircular portion at high frequencies and a linear portion at low frequencies; the charge transfer resistance (Rct) of the electrode was reflected by the semicircular diameter. As can be seen from Figure 5d, the Pt foil electrode has the smallest semicircle diameter, which means that it has the lowest charge transfer resistance. The Rct values of Pt foil, Pt4Ni1-CC, Pt-CC and Ni-CC were obtained by fitting as follows: 2.86 Ω, 3.75 Ω, 5.20 Ω and 31.1 Ω, respectively. The lower charge transfer resistance was conducive to promoting the interfacial electrochemical reaction. Although the Rct value of the PtNi-CC electrode was slightly smaller than that of the Pt foil electrode, it was still better than that of the Pt-CC and Ni-CC electrodes.

To study the electrochemical kinetic process of ammonia-nitrogen detection, the relationship between the scanning rate and peak current was expressed by a CV curve. Figure 5e,f show the CV curves of the Pt4Ni1-CC electrode at different scanning rates from 5 to 150 mV s^−1^ in 1M KOH with 5 mM ammonia. The peak current values were gradually magnified with the increase in scanning rate, and the peak potential moved toward positive displacement. The linear relationship between the peak current value and square root of the scanning rate is shown in Figure 5f. The regression equation is Y = 0.510x + 0.267 (R^2^ = 0.998). It was proved that the electrooxidation process of ammonia is the diffusion control process of Pt4Ni1-CC electrode [35].

### 3.3. Determination of Ammonia

Based on Figure 5a, the Pt4Ni1-CC electrode has a good electrochemical response to ammonia. Therefore, the Pt4Ni1-CC electrode was used for electrochemically detecting ammonia in an electrolyte containing different concentrations of ammonium chloride (1 M KOH with 1 μM~5 mM NH_4_Cl). The CV curve is shown in Figure 6a. When the ammonia concentration was from 1 μM to 5 mM, the peak currents were increased accordingly and maintained a good linear relationship. There were two linear regions from 1 μM to 0.5 mM: Y (mA cm^−2^) = 0.00182x + 0.453 (R^2^ = 0.993) and from 0.5 mM to 5 mM: Y (mA cm^−2^) = 0.000617x + 0.988 (R^2^ = 0.994).

Under the optimal experimental conditions, ammonia was determined on a Pt4Ni1-CC electrode using the DPV method. Figure 6c shows the DPV response of the Pt4Ni1-CC electrode toward ammonia over the concentration range of 0.5 to 500 μM in a 0.1 M KOH electrolyte. As seen from the calibration plot of ammonia in Figure 6d, the peak current on the potential of about −0.35 V increased linearly with the ammonia concentration. Two linear regression equations, respectively, were Y (μA cm^−2^) = 7.83x (μM) + 1.314 from 0.5 μM to 150 μM and Y (μA cm^−2^) = 0.945x (μM) + 1.99 from 150 to 500 μM.

Formulas for estimating the limit of detection (LOD) and limit of quantification (LOQ) are provided below:(3)$$\mathrm{LOD}=\frac{3\sigma }{s}$$
(4)$$\mathrm{LOQ}=\frac{10\sigma }{s}$$
where σ is the standard deviation of blank solution (1M KOH) and *s* is the slope of the calibration plot and also the sensitivity. The sensitivity was calculated at 7.42 μA μm^−1^ by the lower range region, and the LOD and LOQ for ammonia were calculated as 24 nM and 80 nM, respectively.

Table 2 shows some reported electrochemical sensors. Compared with these sensors, the Pt4Ni1-CC sensor had the recognized comprehensive performance of linear range, sensitivity and LOD, which indicated the Pt4Ni1-CC sensor had application potential.

### 3.4. Comprehensive Performance Test

Reusability, reproducibility and stability were essential qualities for electrochemical sensors. The ammonia response of Pt4Ni1-CC was tested by the DPV method, and the actual curves are shown in the Figure 7. We added a concentration of 100 μM interference ions (Na^+^, K^+^, F^−^, NO^2−^, SO_4_^2−^, HCO_3_^−^, Cl^−^ and CO_3_^2−^) into the ammonia-containing electrolyte and tested the selectivity and anti-interference capacity of the Pt4Ni1-CC electrode. It can be seen in Figure 7a,b that the fluorine ion had a big influence on the detection of ammonia, which may be due to the strong corrosion of the fluorine ion to Pt-based nanomaterials [38]. Therefore, the electrode should be avoided in a higher concentration of fluorine-containing ion solution. In addition, in the absence of fluoride ions, other interfering ions almost have no obvious effect on ammonia detection, indicating that the Pt4Ni1-CC electrode had good anti-interference performance.

As seen in Figure 7c, we added 200 μM ammonia solution in a 1 M KOH test electrolyte, and the PtNi-CC electrode was energized eight times. We recorded the peak current curves, and the relative standard deviation (RSD) was calculated to 1.85%. It can be seen in the Figure 7c inset that there is no significant change in the DPV curve after repeating the test eight times. The Pt4Ni1-CC electrodes show excellent stability and repeatability. In addition, Pt4Ni1-CC electrodes were synthesized seven times by the same conditions. As shown in Figure 7b, the DPV current curves of different electrodes in 1 M KOH electrolyte with 200 μM NH_4_Cl were collected and calculated. The RSD was 3.47%, indicating that Pt4Ni1-CC had good reproducibility.

To evaluate the practical application value of the PtNi-CC electrode, the lake water samples in the campus were collected and tested by the calibration method. Different concentrations of ammonia were added to the sample and measured by DPV, using a commercial Pt foil electrode as a control. Table 3 summarizes the results obtained. It was shown that the PtNi-CC electrode has high precision and can be used for the detection of ammonia in actual samples. The recovery rate is 94.73~103.15%, and the relative standard deviation (RSD) is less than 1.60%. The PtNi-CC electrode had good performance in water with a low ammonia concentration and had the potential of practical application.

## 4. Conclusions

In summary, we proposed the application of PtNi alloy nanomaterials for detecting ammonia nitrogen by the electrochemical method. The alloying of Pt and transition metal Ni showed two obvious superiorities. One was to effectively reduce the use of precious metal Pt. Second, compared with Pt-CC and Pt foil electrodes, the comprehensive electrochemical detection ability of the PtNi-CC electrode was enhanced. The Pt4Ni1-CC electrode had an outstanding sensitivity of 7.83 µA µM^−1^ cm^−2^ and LOD of 24 nM in the detection range between 0.5 and 150 μM. When the ammonia concentration ranges from 150 to 500 μM, the sensitivity of the Pt4Ni1-CC electrode was 0.945 µA µM^−1^ cm^−2^. The self-supporting Pt4Ni1-CC electrode had excellent selectivity, anti-interference, reproducibility and repeatability. In summary, the Pt4Ni1-CC electrode has a satisfactory application potential in the sensing of ammonia nitrogen in water. It is further proved that the alloying of the Pt and Ni atom is one of the effective strategies for preparing high-performance ammonia-nitrogen electrochemical sensors.

## Figures and Tables

**Figure 1 sensors-24-00387-f001:**
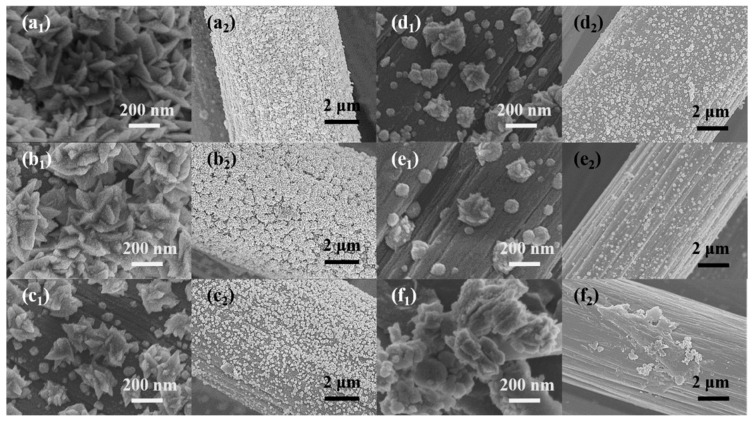
The SEM images of (**a_1_**,**a_2_**) Pt-CC, (**b_1_**,**b_2_**) Pt4Ni1-CC, (**c_1_**,**c_2_**) Pt3Ni2-CC, (**d_1_**,**d_2_**) Pt2Ni3-CC, (**e_1_**,**e_2_**) Pt1Ni4-CC and (**f_1_**,**f_2_**) Ni-CC.

**Figure 2 sensors-24-00387-f002:**
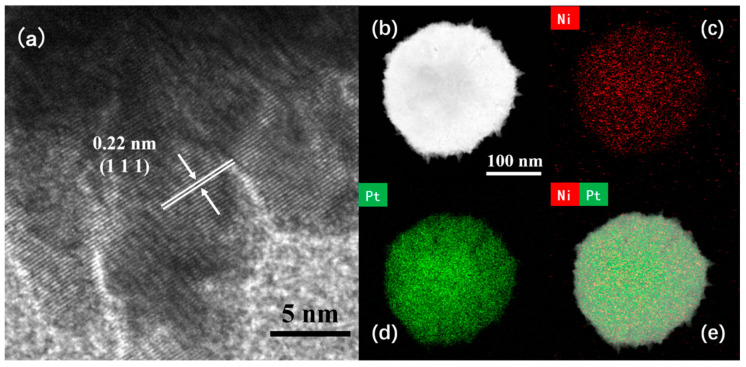
(**a**) HR-TEM and (**b**) HAADF-STEM images of Pt4Ni1-C; (**c**–**e**) elemental mapping images.

**Figure 3 sensors-24-00387-f003:**
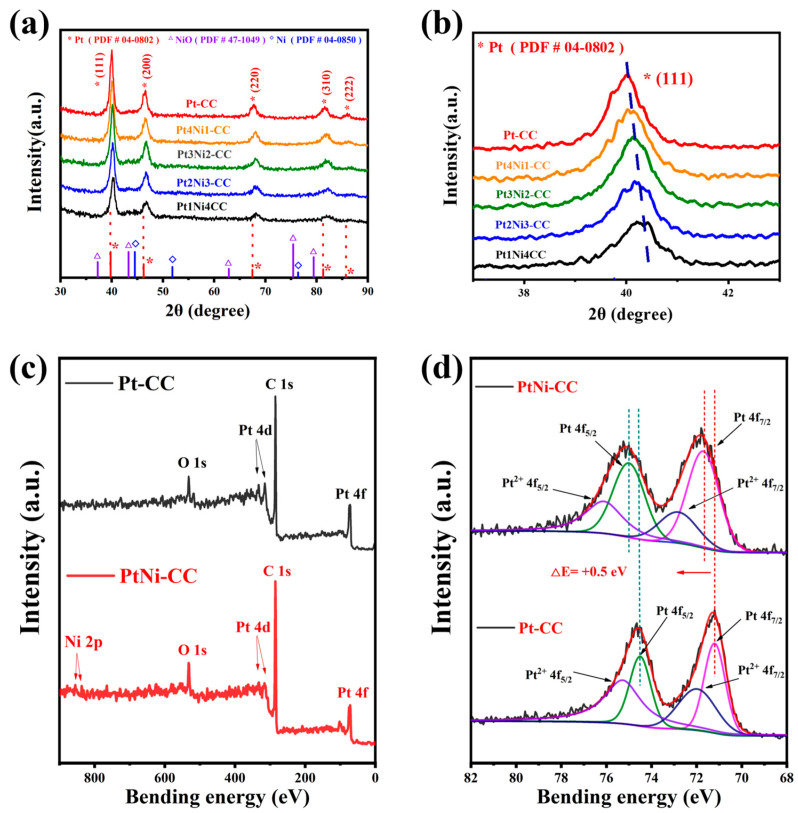
(**a**) XRD spectra of Pt-CC, Pt4Ni1-CC, Pt3Ni2-CC, Pt2Ni3-CC, and Pt1Ni4-CC, (**b**) locally enlarged image; XPS full scan survey spectra and high-resolution XPS spectra of Pt-CC and Pt4Ni1-CC: (**c**) survey, (**d**) Pt 4f. The black solid lines were the original data curves, and the red solid line were fitted curves; the dotted line showed the positions of the binding energies of each major peaks.

**Figure 4 sensors-24-00387-f004:**
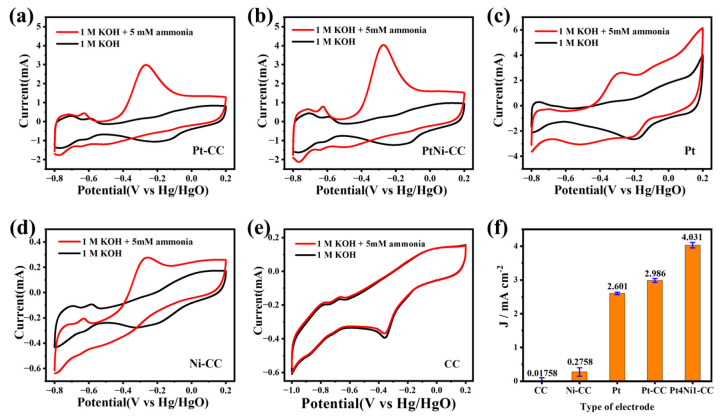
CV responses of (**a**) Pt-CC, (**b**) Pt4Ni1-CC, (**c**) 1 × 1 cm^2^ commercial Pt foil, (**d**) Ni-CC and (**e**) pure CC electrode and (**f**) the value of the current density fitting diagram for different electrodes toward 5 mM ammonia at different concentrations in 0.1 M KOH solution.

**Figure 5 sensors-24-00387-f005:**
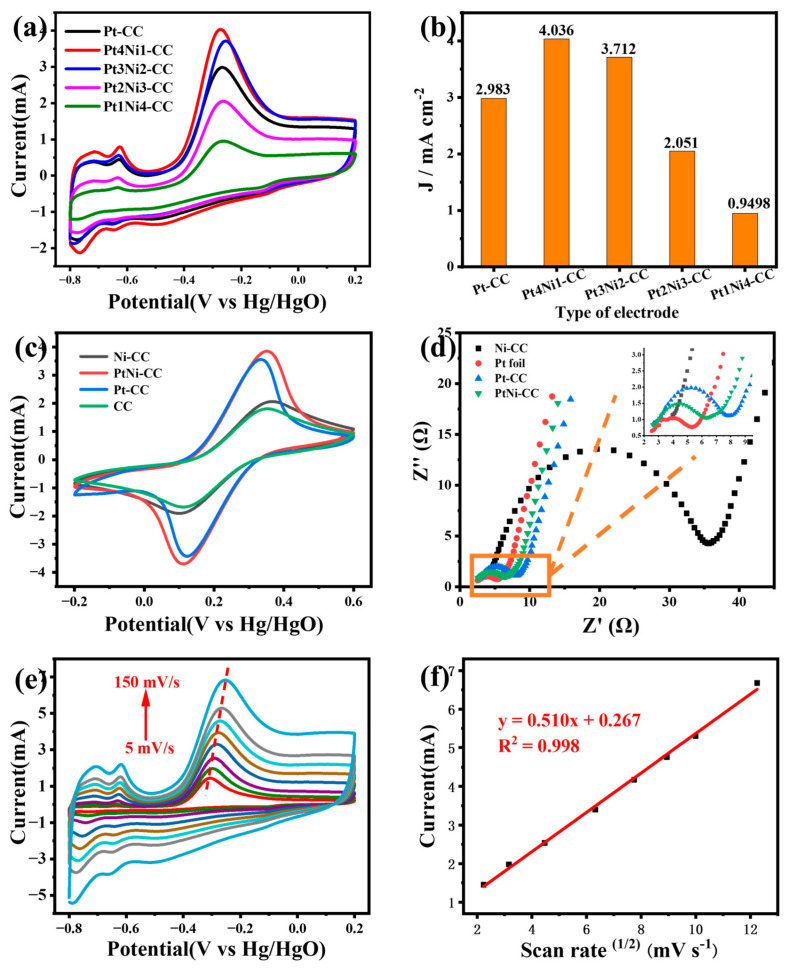
(**a**) CV curves of Pt-CC, Pt4Ni1-CC, Pt3Ni2-CC, Pt2Ni3-CC and Pt1Ni4-CC in 1 M KOH with 5 mM NH_4_Cl solution and (**b**) the value of the current density-fitting diagram for different electrodes. (**c**) CV curves and (**d**) electrochemical impedance spectra of different electrodes in 0.1 M KCl solution containing 5.0 mM [Fe(CN)_6_]^3−/4−^. (**e**) The CV curves of Pt4Ni1-CC electrode with different scan rates from 5 to 150 mV s^−1^ in 1 M KOH and 5 mM NH_4_Cl solution and (**f**) calibration curve of peak current vs. the square root of scan rates.

**Figure 6 sensors-24-00387-f006:**
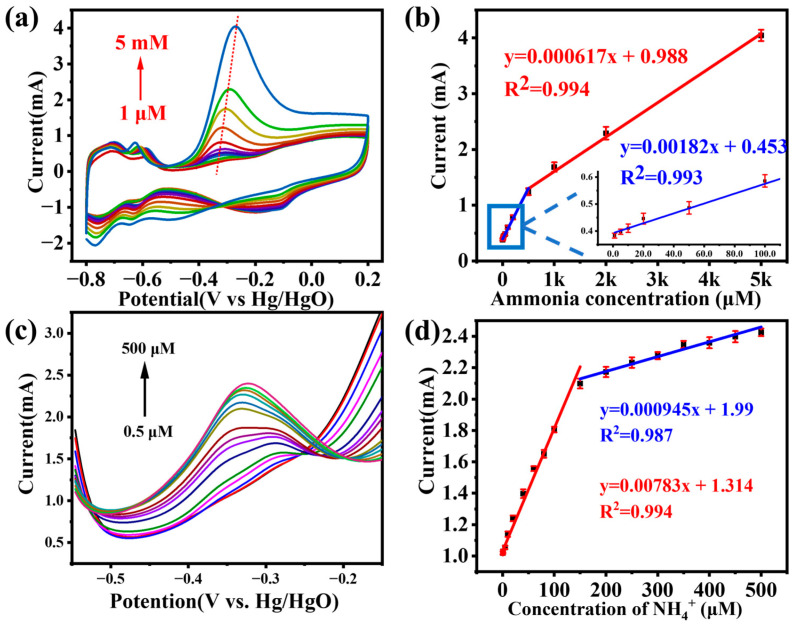
(**a**) CV curves of Pt4Ni1-CC in the presence of different concentrations of ammonia and (**b**) corresponding calibration curves; (**c**) DPV curves of the Pt4Ni1-CC electrode to different concentration of ammonia; (**d**) the calibration curve of the peak current values vs. the concentration of NH_4_Cl from 0.5 to 500 μM.

**Figure 7 sensors-24-00387-f007:**
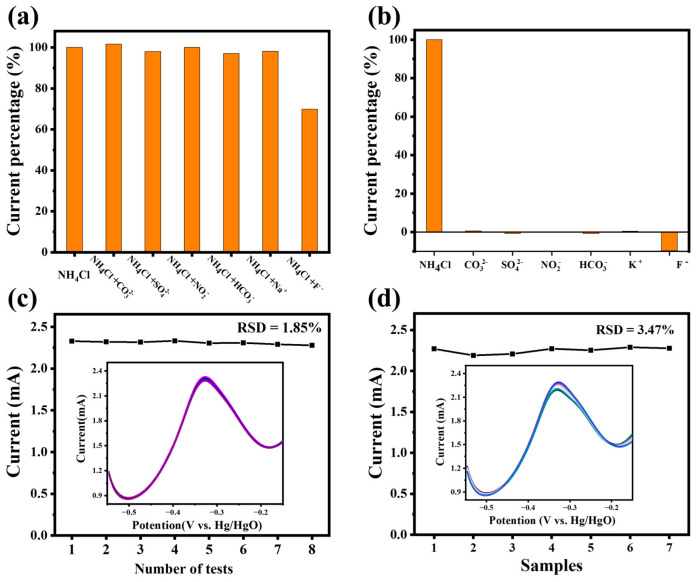
(**a**) Anti-interference test of Pt4Ni1-CC electrode; (**b**) selectivity test of the Pt4Ni1-CC electrode to ammonia and other interferences; (**c**) DPV response current values of the Pt4Ni1-CC electrode for testing eight times in the electrolyte with 200 μM NH_4_Cl, the inset shows the DPV curves; (**d**) DPV response current values of 7 different Pt4Ni1-CC electrodes in the electrolyte with 200 μM NH_4_Cl; the inset shows the DPV response curves.

**Table 1 sensors-24-00387-t001:** Binding energy and relative content of Pt 4f XPS curve fitting for Pt and PtNi catalysts.

Species	$\mathrm{Pt}\;{4f}_{7/2}$ (eV)	$\mathrm{Pt}\;{4f}_{5/2}$ (eV)	Relative Concentrations (%)
XPS Pt Species of Pt-CC			
Pt 0	71.20	74.50	45.67%
Pt 2^+^	72.00	75.30	54.33%
XPS Pt Species of PtNi-CC			
Pt 0	71.70	75.00	62.50%
Pt 2^+^	72.81	76.11	37.50%

**Table 2 sensors-24-00387-t002:** The method of ammonia-nitrogen detection and its electrosensitive performance reported in the literature.

Electrode Type	Detection Range (μM)	Sensitivity (μA μM^−1^)	LOD	Method	Ref
/	20–35 ppt	939	4 ppt	salicylate method	[1]
/	0.85–5 mg/L	/	0.6 mg/L	Nessler	[36]
SPE	0.025–0.50 mg/L	/	0.010 mg/L	SPE-indothymol	[36]
Fluorescent probe	0–5	/	3.5 nM	fluorimetry	[7]
Cu NPs/CC	5–9425	0.0062	1.25 μM	i-t	[15]
Ag/Fe_2_O_3_/TNT	0.5–134	1876	0.18 μM	i-t	[24]
Pt-Ni(OH)_2_	0.05–600	0.191	39.2 nM	DPV	[37]
Pt-Ni(OH)_2_-NF	5–500	12.27	2.74 μM	DPV	[38]
Pt-Ag/PPy	0.05–50	8.9	37 nM	CV/LSV	[39]
Pt7Cu1	0.5–500	9.4	8.6 nM	DPV	[12]
Pt4Ni1-CC	150–500 0.5–150	0.945 7.83	24 nM	CV/DPV	This work

**Table 3 sensors-24-00387-t003:** Determination of ammonia in real samples (lake water).

Sample	Initial (μM)	Added (μM)	Found (μM)	Recovery (%)	RSD (%, *n* = 3)
Pt4Ni1-CC	3.56	10	13.03	94.73	1.60
		30	34.50	103.15	1.23
Commercial Pt foil	1.40	10	8.72	73.18	0.70
		30	30.57	97.23	1.14

## Data Availability

Data are contained within the article.

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
