# Peer review of "Enhanced Sensitivity of Electrochemical Sensors for Ammonia-Nitrogen via In-Situ Synthesis PtNi Nanoleaves on Carbon Cloth"

_sensors, 2024, doi:10.3390/s24020387_

Round 1

Reviewer 1 Report

Comments and Suggestions for Authors

This paper proposed a a self-supported electrochemical electrode for the detection of ammonia nitrogen was obtained by elecrodeposition of PtNi alloy nanoleaves on carbon cloth (PtNi-CC). The method showed that the PtNi-CC electrode exhibited enhanced detection performance with the wide linear range from 0.5 ~ 500 µM, high sensitivity (7.42 µA µM-1 cm-2) and lower detection limit (24 nM). After reviewing this paper, it is deemed suitable for publication in this journal after modification the following minor revision:

(1) For the modified electrode, the electroactive area is a key parameter. The author should add the relevant experiment and calculation.

(2) The introduction section is too simple and limited, and relevant literature of transition metal and precious metal used for sensitive material should be summarized.

(3) text formatting errors in the manuscript. Please check out and modify.

(4) The style of the references should be further proof and revised.

(5) In figure 6a, the author should describe clearly.

Comments on the Quality of English Language

This paper proposed a a self-supported electrochemical electrode for the detection of ammonia nitrogen was obtained by elecrodeposition of PtNi alloy nanoleaves on carbon cloth (PtNi-CC). The method showed that the PtNi-CC electrode exhibited enhanced detection performance with the wide linear range from 0.5 ~ 500 µM, high sensitivity (7.42 µA µM-1 cm-2) and lower detection limit (24 nM). After reviewing this paper, it is deemed suitable for publication in this journal after modification the following minor revision:

(1) For the modified electrode, the electroactive area is a key parameter. The author should add the relevant experiment and calculation.

(2) The introduction section is too simple and limited, and relevant literature of transition metal and precious metal used for sensitive material should be summarized.

(3) text formatting errors in the manuscript. Please check out and modify.

(4) The style of the references should be further proof and revised.

(5) In figure 6a, the author should describe clearly.

Author Response

A detailed reply to the reviewers’ comments

The authors would like to express their sincere appreciation to the editor and the reviewers for their diligent evaluation of our manuscript. We are truly grateful for their valuable comments and suggestions, which have greatly enhanced the quality of our research. The manuscript has undergone thorough revisions, encompassing all the suggestions and corrections put forth by the editor and the reviewers. In response to the reviewers' insightful feedback, we have provided elucidations and clarifications where necessary. The revised manuscript has been meticulously proofread and refined, with all modifications and additions clearly highlighted for easy reference. Once again, we would like to extend our heartfelt gratitude to the editor and the reviewers for their invaluable contributions to this work. A detailed reply to the reviewers’ comments is provided as follows:

Reviewers’ comments:

Reviewer #1: This paper proposed a a self-supported electrochemical electrode for the detection of ammonia nitrogen was obtained by elecrodeposition of PtNi alloy nanoleaves on carbon cloth (PtNi-CC). The method showed that the PtNi-CC electrode exhibited enhanced detection performance with the wide linear range from 0.5 ~ 500 µM, high sensitivity (7.42 µA µM-1 cm-2) and lower detection limit (24 nM). After reviewing this paper, it is deemed suitable for publication in this journal after modification the following minor revision:

Authors’ response:

Thank you very much for your comments and meticulous work. Your suggestions are all very meaningful and helpful for this work. The authors are highly appreciated for your work.

Comment 1:

For the modified electrode, the electroactive area is a key parameter. The author should add the relevant experiment and calculation.

Authors’ response:

Thanks for your precious advice. According to your suggestions, we have added the data of the electrochemical active area and electrochemical impedance spectroscopy in the manuscript, to better study the significance of the electron transport kinetics and interfacial electrochemical properties of the modified electrode. Specific changes are as follows:

“It is very important to study the electron transport kinetics and interfacial elec-trochemical properties of the modified electrode. As described in Figure 5c, all the electrodes revealed reversible redox peaks (I PtNi-CC> I Pt-CC> I Ni-CC> I CC). The peak current of PtNi-CC was significantly higher than that of other electrodes. This showed that the electrode obtained larger active specific surface area and higher conductivity by electrodepositing PtNi nanoleaves on the bare CC surface.

Figure 5d depicted the Nyquist plots of PtNi-CC, Pt-CC, PtNi-CC, PtNi-CC and bare Pt foil electrodes, and the illustration was the partial enlarged drawing. The Nyquist plots consisted of a semicircular portion at high frequencies and a linear portion at low frequencies, the charge transfer resistance (Rct) of the electrode was re-flected by the semicircular diameter. As can be seen from Figure 5d, the Pt foil electrode has the smallest semicircle diameter, which means that it has the lowest charge transfer resistance. The Rct values of Pt foil, Pt4Ni1-CC, Pt-CC and Ni-CC were obtained by fitting, as follows: 2.86 Ω, 3.75 Ω, 5.20 Ω and 31.1 Ω, respectively. The lower charge transfer resistance was conducive to promoting the interfacial electrochemical reaction. Although the Rct value of the PtNi-CC electrode was slightly smaller than that of the Pt foil electrode, it was still better than that of the Pt-CC and Ni-CC electrodes.”

Comment 2:

The introduction section is too simple and limited, and relevant literature of transition metal and precious metal used for sensitive material should be summarized.

Authors’ response:

       Thank you for your valuable advice. Based on your comments, we have reviewed the research on sensitive electrodes of precious metal and transition metal materials in the introduction. The contents are as follows:

“Ammonia sensitive electrode materials mainly include precious metal nanomaterials, transition metals and their oxides and conductive polymers. Researchers have conducted many studies on these sensitive materials. G. Novell-Leruth et al. calculated the adsorption characteristics of Pt(100) and Pt(111) crystal faces for ammonia by using DFT theory. Kazuki Endo et al. prepared Pt-Ir and Pt-Cu binary alloys as electrocatalysts for ammonia oxidation. They proved that a synergistic interaction between Pt and Ir can enhance the ammonia oxidation activity. Yang et al. implemented 3D Rosett-like Cu nanoparticles with sensitive detection of ammonia nitrogen. Junhua Jiang et al. studied the effect of Zn modification on the catalytic activity of PtIr and Pt for ammonia electrooxidation. In many studied species, noble metal nanomaterials, especially platinum, show excellent sensitivity to ammonia nitrogen. Although transition metals can also realize better ammonia sensitive properties, they were often used as modified materials and combined with precious metal materials to obtain better performance sensitive electrode materials.”

“At present, transition metal alloying was an effective way to solve these problems, and this method has shown excellent results in different kinds of fields, like hydrogen evolution reaction (HER), oxygen reduction reaction (ORR), ammonia oxidation reaction (AOR) [16–18]. Zhang et al. constructed surface-structure tailoring of dendritic PtCo nanowires for efficient oxygen reduction reaction; Tran et al. Graphene coated PtNi nanosponge enhanced oxygen reduction performance.”

Comment 3:

       Text formatting errors in the manuscript. Please check out and modify.

Authors’ response:

Thank you for your valuable advice. We carefully corrected the formatting errors in the manuscript and highlighted the changes in red.

Comment 4:

       The style of the references should be further proof and revised.

Authors’ response:

Thank you for your valuable advice. We checked the format of the references and re-inserted them with Zotero software.

Comment 5:

       In figure 6a, the author should describe clearly.

Authors’ response:

Thank you for your valuable advice. We have redescribed Figure 6(a-b) (Figure 6c-d in the manuscript now) as follows.

“Under the optimal experimental conditions, ammonia was determined on a Pt4Ni1-CC electrode using DPV method. Figure 6c shown the DPV response of the Pt4Ni1-CC electrode toward ammonia over the concentration range of 0.5 to 500 μM in 0.1 M KOH electrolyte. As seen from the calibration plot of ammonia in figure 6d, the peak current on the potential about -0.35V increased linearly with the ammonia concentration.  Two linear regression equations respectively were Y (μA cm-2) = 7.83 x (μM) + 1.314 from 0.5 μM to 150 μM and Y (μA cm-2) = 0.945 x (μM) + 1.99 from 150 μM to 500 μM.”

Reviewer 2 Report

Comments and Suggestions for Authors

In a present state the manuscript submitted is adequate and to be recommended for publication with major correction. 

Listed of the corrections and question:

11.     Results

i.                 Figure 4 (f) and 5 (a): The value of current must be measured not a point of higher peak.

ii.                Figure 4 (f),  5 (a),  6(b) and 7 :  Should insert error bar

iii.               Refer to Figure 5 (d) : The sensor has two (2) linear range and should be stated in abstract.

iv.              Refer to Table 2 : [14] LOD = 39.2 is wrong because detection range 0.05 – 600. LOD must be not more 0.05. Please check the article carefully.

v.                Some of electrochemical performances of sensor (i.e. EIS) must be study such as impedance spectroscopy which from that data such as surface area of sensor, kinetic study etc will strengthen the results.

22.     Real samples analysis should be study and compared (validated) with commercial instrument.

33.     Check all the spelling error. i.e : 2.1,  line 89 : Ammonium fluoride should be ammonium fluoride

Reviewer 3 Report

Comments and Suggestions for Authors

The work entitled "Enhanced sensitivity of electrochemical sensors for ammonium-nitrogen via in-situ synthesis PtNi nanoleaves on carbon cloth" is devoted to the synthesis and testing of a series of new sensors based on PtNi. It is unnecessary to recall the relevance of rapid and reproducible detection of ammonia in various objects. In addition, the self-supported greatly increases the durability of such sensors. The work has great potential for acceptance to Sensors after correcting the following comments.

1.        The authors are recommended to compare the studied sensor samples (in addition to those shown in Table 2) primarily with industrial technological samples that have long been widely used for the detection of ammonium and ammonia in order to show the advantages of new samples.

2.        Line 91 and other places. It is required to check the spelling of nickel chloride.

3.        It is recommended to divide the large and important Section 3 into logical subsections, now there is only subsection 3.1, which clearly does not correspond to the entire large volume of work performed by the authors.

4.        Line 173. After the name of the sample, it is better and more logical to put a colon, not a comma, as it is now.

5.        Figure 4f. Which of the samples Pt?Ni-CC is indicated in the last column, it should be clarified.

6.        Line 228. Where are x and y used in the equations? What do they relate to?

7.        Line 236. How can improved desorption contribute to improved ammonia activity? Usually, on the contrary, increased adsorption should increase the detection limit. The authors should explain this statement.

8.        Line 241. In Figures 5a, b, there is ammonium everywhere. What kind of absence are we talking about?

9.        Line 249. It is not at all clear what kind of deposition we are talking about? What did the authors mean here?

10.     Figure 5d insertion. The values are not visible at all, especially along the axes.

11.     Figure 5a. The names on the x-axis are poorly and not clearly visible.

12.     The names of the journals in the references are not always correctly abbreviated.

Author Response

A detailed reply to the reviewers’ comments

The authors would like to express their sincere appreciation to the editor and the reviewers for their diligent evaluation of our manuscript. We are truly grateful for their valuable comments and suggestions, which have greatly enhanced the quality of our research. The manuscript has undergone thorough revisions, encompassing all the suggestions and corrections put forth by the editor and the reviewers. In response to the reviewers' insightful feedback, we have provided elucidations and clarifications where necessary. The revised manuscript has been meticulously proofread and refined, with all modifications and additions clearly highlighted for easy reference. Once again, we would like to extend our heartfelt gratitude to the editor and the reviewers for their invaluable contributions to this work. A detailed reply to the reviewers’ comments is provided as follows:

Reviewers’ comments:

Reviewer #3: The work entitled "Enhanced sensitivity of electrochemical sensors for ammonium-nitrogen via in-situ synthesis PtNi nanoleaves on carbon cloth" is devoted to the synthesis and testing of a series of new sensors based on PtNi. It is unnecessary to recall the relevance of rapid and reproducible detection of ammonia in various objects. In addition, the self-supported greatly increases the durability of such sensors. The work has great potential for acceptance to Sensors after correcting the following comments.

Authors’ response:

Thank you very much for your comments and meticulous work. Your suggestions are all very meaningful and helpful for this work. The authors are highly appreciated for your work.

Comment 1:

The authors are recommended to compare the studied sensor samples (in addition to those shown in Table 2) primarily with industrial technological samples that have long been widely used for the detection of ammonium and ammonia in order to show the advantages of new samples.

Authors’ response:

Thanks for your precious advice. In Table 2, we listed four commonly used ammonia nitrogen detection methods for research and comparison with our sensitive electrodes.

Table 2. The method of ammonia nitrogen detection and its electrosensitive performance reported in the literature.

Electrode type

Detection range (μM)

Sensitivity (μA μM-1)

LOD

Method

Ref

/

20 – 35 ppt

939

4 ppt

salicylate method

[1]

/

0.85 – 5 mg/L

/

0.6 mg/L

Nessler

[36]

SPE

0.025 – 0.50 mg/L

/

0.010 mg/L

SPE-indothymol

[36]

fluorescent probe

0 - 5

/

3.5 nM

fluorimetry

[7]

Comment 2:

             Line 91 and other places. It is required to check the spelling of nickel chloride.

Authors’ response:

Thank you for your careful examination and precious advice. We checked the manuscript for spelling errors and formatting errors. We have corrected the errors and marked them in red. We will try to avoid such problems in the future.

Comment 3:

It is recommended to divide the large and important Section 3 into logical subsections, now there is only subsection 3.1, which clearly does not correspond to the entire large volume of work performed by the authors.

Authors’ response:

Thanks for your precious advice. We divide section 3(Results and Discussion) into four logical subsections, which are:

3.1. Material Characterization

3.2. Study on electrochemical sensitivity characteristics

3.3. Determination of ammonia

3.4. Comprehensive performance test

Comment 4:

Line 173. After the name of the sample, it is better and more logical to put a colon, not a comma, as it is now.

Authors’ response:

Thank you for pointing out the formatting error, we have corrected it. We will avoid such problems in the future.

Comment 5:

 Figure 4f. Which of the samples PtNi-CC is indicated in the last column, it should be clarified.

Authors’ response:

Thanks for your precious advice. In Figure 4f, we have marked the name of the sample at the corresponding X-axis position at the bottom of each column.

Comment 6:

Line 228. Where are x and y used in the equations? What do they relate to?

Authors’ response:

Thank you for your careful examination. This line in the manuscript should have been removed. We are sorry for our mistake. We have deleted it.

Comment 7:

Line 236. How can improved desorption contribute to improved ammonia activity? Usually, on the contrary, increased adsorption should increase the detection limit. The authors should explain this statement.

Authors’ response:

Thank you for pointing out the problem for us. According to the literature reports, the DFT theoretical calculation results showed that PtNi nanomaterials can promote the desorption of NH4+ from the catalyst surface. We quoted that view. But we realized that we didn't have enough data to support it. Therefore, we have decided to delete this sentence. The revised description is as follows:

“According to the above XRD patterns, the crystal plane spacing of Pt decreased after the formation of PtNi alloy, which was conducive to improve the electrochemical activity of ammonia in the electrooxidation process and improved the response current.”

Comment 8:

Line 241. In Figures 5a, b, there is ammonium everywhere. What kind of absence are we talking about?

Authors’ response:

Thanks for your precious advice. Here is our misrepresentation, the test was indeed carried out in an electrolyte containing 5 mM ammonia. We have corrected this. The corrected description is as follows:

“The effect of different element ratio on PtNi-CC electrode was evaluated. Pt-CC, Pt4Ni1-CC, Pt3Ni2-CC, Pt2Ni3-CC, and Pt1Ni4-CC electrode were tested in the electrolyte with 5 mM ammonia (Fig. 5a-b).”

Comment 9:

Line 249. It is not at all clear what kind of deposition we are talking about? What did the authors mean here?

Authors’ response:

Thanks for your precious advice. We apologize for not making our point clearly. In the SEM and XRD characterization, we can see from the data that too many Ni atoms will affect the crystal structure and the morphology of the onlookers, so the electrochemical response of ammonia may be affected. Therefore, here we try to confirm the above view with the data in Figure 5a-b. However, our part of the data does not seem to support this view effectively. Based on the unprecise and unclear description presented here, we have decided to delete this sentence so as not to misunderstand the reader.

Comment 10:

Figure 5d insertion. The values are not visible at all, especially along the axes.

Authors’ response:

Thanks for your precious advice. We have re-drawn Figure 5d (now is Figure 6b) and the insertion to show as much valuable information as possible. In the insertion, the response current values in the range of 0 μM ~ 100 μM after magnification can be seen to have a high linear fit.

Comment 11:

Figure 5a. The names on the x-axis are poorly and not clearly visible.

Authors’ response:

Thanks for your precious advice. We have re-drawn Figure 5a (now is Figure 6a) and other graphs that have the same problem. We magnified the names and numbers in the figures to make them as clear as possible.

Comment 12:

 The names of the journals in the references are not always correctly abbreviated.

Authors’ response:

Thanks for your precious advice. We re-edited the references with Zoreto software to ensure they meet the requirements of sensors.

Round 2

Reviewer 2 Report

Comments and Suggestions for Authors

Accepted.

Author Response

A detailed reply to the reviewers’ comments

The authors would like to express their sincere appreciation to the editor and the reviewers for their diligent evaluation of our manuscript.   We are truly grateful for their valuable comments and suggestions, which have greatly enhanced the quality of our research. The manuscript has undergone thorough revisions, encompassing all the suggestions and corrections put forth by the editor and the reviewers. In response to the reviewers' insightful feedback, we have provided elucidations and clarifications where necessary. The revised manuscript has been meticulously proofread and refined, with all modifications and additions clearly highlighted for easy reference. Once again, we would like to extend our heartfelt gratitude to the editor and the reviewers for their invaluable contributions to this work.

Reviewer 3 Report

Comments and Suggestions for Authors

After the author's correction of the comments, the manuscript is almost ready for acceptance. Only minor corrections remain now:

Comment 2. Additional comment: Correct spelling of NiCl2 . Please correct it here and elsewhere.

Comment 5. Additional comment: A sample of PtNi-CC is written on Figure 4a. Most likely it is Pt4Ni1-CC? If this is the case, then correct it in the Figure and in the corresponding text too.

Pay attention and be sure to correct the Figure 6c in the axis caption to Potential.

Author Response

The authors would like to express their sincere appreciation to the editor and the reviewers for their diligent evaluation of our manuscript. We are truly grateful for their valuable comments and suggestions, which have greatly enhanced the quality of our research. The manuscript has undergone thorough revisions, encompassing all the suggestions and corrections put forth by the editor and the reviewers. In response to the reviewers' insightful feedback, we have provided elucidations and clarifications where necessary. The revised manuscript has been meticulously proofread and refined, with all modifications and additions clearly highlighted for easy reference. Once again, we would like to extend our heartfelt gratitude to the editor and the reviewers for their invaluable contributions to this work. A detailed reply to the reviewers’ comments is provided as follows:
